# Genome-Wide Identification of the AGC Protein Kinase Gene Family Related to Photosynthesis in Rice (*Oryza sativa*)

**DOI:** 10.3390/ijms232012557

**Published:** 2022-10-19

**Authors:** Yifei Jiang, Xuhui Liu, Mingao Zhou, Jian Yang, Simin Ke, Yangsheng Li

**Affiliations:** State Key Laboratory of Hybrid Rice, Key Laboratory for Research and Utilization of Heterosis in Indica Rice of Ministry of Agriculture, College of Life Sciences, Wuhan University, Wuhan 430072, China

**Keywords:** rice, *Oryza* genus, AGC protein kinase, gene family, photosynthesis

## Abstract

The cAMP-dependent protein kinase A, cGMP-dependent protein kinase G and phospholipid-dependent protein kinase C (AGC) perform various functions in plants, involving growth, immunity, apoptosis and stress response. *AGC* gene family is well described in *Arabidopsis*, however, limited information is provided about *AGC* genes in rice, an important cereal crop. This research studied the *AGC* gene family in the AA genome species: *Oryza sativa* ssp. *japonica*, *Oryza sativa* ssp. *indica*, *Oryza nivara*, *Oryza rufipogon*, *Oryza glaberrima*, *Oryza meridionalis*, *Oryza barthii*, *Oryza glumaepatula* and *Oryza longistaminata* were searched and classified into six subfamilies, and it was found that these species have similar numbers of members. The analysis of gene duplication and selection pressure indicated that the *AGC* gene family expanded mainly by segmental or whole genome duplication (WGD), with purifying selection during the long evolutionary period. RNA-seq analysis revealed that *OsAGCs* of subfamily V were specifically highly expressed in leaves, and the expression patterns of these genes were compared with that of photosynthesis-related genes using qRT-PCR, discovered that *OsAGC9*, *OsAGC20*, and *OsAGC22* might participate in photosynthesis. These results provide an informative perspective for exploring the evolutionary of *AGC* gene family and its practical application in rice.

## 1. Introduction

Protein phosphorylation is an extensive strategy used for the regulation of cellular and organismal functions in eukaryotes [1,2,3]. Eukaryotes are rich in protein kinases, a class of enzymes that catalyze protein phosphorylation, and the AGC protein kinase family is one of the seven conserved protein kinase families. AGC protein kinases are a group of serine/threonine protein kinases, including cAMP-dependent protein kinase A (PKA), cGMP-dependent protein kinase G (PKG) and phospholipid-dependent protein kinase C (PKC). The functions of AGC protein kinases have been extensively researched, like participating in the signal transduction of auxin, the regulation of plant immunity, and the modulation of cell growth and apoptosis [4,5].

AGC protein kinases have been studied to varying degrees in different plants, with the most widely reported in *Arabidopsis thaliala* and a few reports in *Solanum lycopersicum*, *Triticum aestivuml*, *Medicago truncatula* and *Porphyra yezoensis*. AGCs were comprehensively identified in the model plant *A. thaliana* and were classified into six subfamilies, AGCs in the PDK1 subfamily bind to some signaling lipids such as phosphatidic acid, PtdIns3P and PtdIns(3,4)P2, and play a fundamental role in signaling processes controlling the pathogen/stress response, polar auxin transport and development [6,7,8]. AGCs in subfamily AGCVI regulate the phosphorylation of MRFs and the TOR signaling pathway, which can also inhibit cell proliferation and be involved in adaptation to cold or high salt conditions [9,10]. AGCs in the subfamily AGCVII are engaged in the regulation of cell morphology, exit from mitosis and cell division [11]. AGCs in subfamily AGC other modulate the growth of the root tip and cell tip [12]. PIDs in subfamily AGC VIIIa regulate the polar transport of auxin, AGC1-5 affects the ROP signaling pathway to determine the apical growth of root hairs, AGC1-5 and AGC1-7 are involved in the polarized growth of pollen tubes [13,14,15,16,17,18]. UNICORN in subfamily AGCVIIIb regulates the development of the planar ovule integument and restricts the growth of stamen filaments, petals and cotyledons, phototropins (AtPHOT1 and AtPHOT2) can mediate phototropism, chloroplast movement and leaf flattening in *A. thaliala*, phototropins were also found to regulate dark-induced leaf senescence [19,20,21,22,23]. The extension of the T-loop of AGC kinase Adi3 guides nuclear localization and thus suppresses cell death, and there is a potential link between the inhibition of PDK1 activity and cell death triggered by reactive oxygen NO in *S. lycopersicum* [24,25]. The AGC kinase TaAGC1 in *T. aestivuml*, plays an active role in immunity to the necrotrophic pathogen Rhizoctonia cerealis by regulating the expression of ROS-related and defense-related genes [26]. The AGC kinase MtIRE in *M. truncatula* exhibits special expression in the rhizome invasion zone [27]. Mitosis in *P. yezoensis* may be governed by the PI3K-AGC signaling pathway [28].

Rice (*Oryza sativa*) is a staple food for more than half of the world’s population [29,30]. Some studies on AGC kinases have also reported that the OsPID kinase controls the polar transport of auxin, mediates stigma and ovule initiation and co-regulates spikelet or flower development in the presence of interactions with OsNPY [31,32,33]. OsPdk1 kinase is responsible for basal disease resistance in rice via a phosphorylation cascade of OsOxi1-OsPti1a in rice [34,35]. Two AGC kinases (OsD6PKL3.6/3.7) were found to regulate the formation of pollen aperture possibly through interactions with an FLA family protein DEAP1 [36]. However, there are more AGC kinases whose functions are unclear in rice. Based on this background, whole genome-wide identification of the AGC protein kinase family was performed, which may lay a foundation for the understanding of the evolutionary relationship and functional exploration of AGC kinases in rice.

## 2. Results

### 2.1. Identification of AGC Gene Family Members in Oryza Genus

To identify all members of the *AGC* gene family in AA genome rice, the amino acid sequences of 20 AGC proteins from *Arabidopsis* were used as queries for Hidden Markov model construction and BLASTp search. Moreover, 26, 27, 26, 27, 25, 26, 26, 28, 29 AGCs were identified in *O. sativa*, *japonica*, *O. sativa*, *indica*, *O. nivara*, *O. rufipogon*, *O. glaberrima*, *O. meridionalis*, *O. barthii*, *O. glumaepatula*, *O. longistaminata*, respectively (Appendix A). The protein sequences of AA genome rice and *Arabidopsis* were used to construct the phylogenetic tree, showing that AGCs were divided into six subfamilies. Obviously, subfamily V possessed the largest number of members that might play a major role in the expansion of the *AGC* gene family, and the remaining subfamilies possessed fewer members (Figure 1).

### 2.2. Characterization and Phylogenetic Relation of OsAGC Gene Family Members in Rice

*OsAGC* genes were distributed on all chromosomes, with the highest number of genes identified on chromosome 12, and they were named according to their location information on the chromosome. Some characteristics of the proteins encoded by these *OsAGC* genes were predicted, with amino acid residue numbers ranging from 338 (*OsAGC23*) to 1267 (*OsAGC8*), molecular weight from 36,565 to 138,833, and isoelectric point from 5.32 to 9.50. The subcellular localization sites of most proteins were located in the nucleus, while others were located in the endomembrane, chloroplast, plasma membrane, and chloroplast outer membrane (Table 1). To better understand the developmental relationship among *OsAGC* gene family members, a phylogenetic tree was constructed using the alignment of AGC proteins sequences from rice and *Arabidopsis* (Figure 2a). According to the cluster analysis, subfamily I and subfamily V contain three members, subfamily III and subfamily IV contain two members, subfamily VI contains fifteen members, and subfamily II has only one member.

### 2.3. Analysis of Gene Structures and Conserved Motifs

The analysis of gene structures and conserved motifs facilitated a better understanding of the *OsAGC* gene family members. Ten representative motifs were selected to explore the association between *OsAGC* gene family members and these motifs’ information was shown in Appendix A. Overall, all members shared more than half of the target motifs, and the distribution of motifs was similar within the same subfamily. Interestingly, subfamily V possessed all kinds of motifs, suggesting that these motifs may be involved in their standard functions and the duplicated motifs identified may perform significant functions. In addition, the motifs of the members of subfamily III and IV were not different in number and type, indicating that these members may be involved in executing the same function. Some members of subfamily I, II and VI held a specific type of motifs, showing that they may be engaged in various physiological processes (Figure 2b). As expected, genes of the same subfamily shared similar exon/intron structures. Genes of subfamily I possessed similar sequence length, subfamily II, III, IV and V held relatively abundant exons/introns, and most genes of the largest subfamily VI owned introns, UTRs, and two CDS regions (Figure 2c).

In summary, the phenomenon that members in different subfamilies owned variable gene structures and conserved motifs further supported the phylogenetic clustering of *OsAGC* gene family.

### 2.4. Collinearity Analysis of AGC Genes in Oryza Genus

Gene duplication events are critical for gene family formation, and the analysis of duplication events was conducted using the Multiple Covariance Scanning Toolkit (MCScanX) method, and the distributions of collinear gene pairs on chromosomes were shown by TBtools software. In total, 6, 8, 8, 9, 7, 5, 8 and 9 collinear gene pairs were found in *O. sativa*, *japonica*, *O. sativa*, *indica*, *O. nivara*, *O. rufipogon*, *O. glaberrima*, *O. meridionalis*, *O. barthii*, and *O. glumaepatula* separately (Appendix A). The duplication types of all collinear gene pairs were segmental/WGD duplication, and the gene pairs possessed similar positions existing in at least one genome, showing that the expansion mode of the *AGC* gene family in various rice species was approximate (Figure 3; Table 2).

The Ka/Ks values of collinear gene pairs were used to evaluate the selection pressure of duplicated genes. In common, Ka/Ks < 1, =1 and >1 indicated that the genes have undergone purifying, neutral and positive selection, correspondingly. Ka/Ks values of all collinear gene pairs in the eight rice varieties were observed to be less than 1, demonstrating that these genes experienced strong purifying selection in the process of evolution. Besides this, the duplication events of 60 gene pairs were estimated to occur ranging from 9.34 to 137.23 Mya (Table 2).

### 2.5. Gene–CDS–Haplotype (gcHap) Analysis of AGC Genes in 3010 Rice Genomes (3KRG)

In the course of long-term growth and development in plants, genes evolved a rich variety of haplotypes. The *AGC* genes carried at least 18 gcHaps in 3KRG, and the mean gcHap number (gcHapN) value, number of major haplotypes (≥30 rice accessions) and Shannon’s equitability (*E_H_*) of them were 389.58 ± 624.93, 5.23 ± 2.89 and 0.30 ± 0.24, respectively. There was little difference in the mean gcHapN value of *Xian/indica* (*XI*), *Geng/japonica* (*GJ*), *Aus*, *Basmati* (*Bas*), and *Admixtures* (*Adm*) populations. The mean gcHapN value was highest in *XI* (252.35 ± 407.23), followed by *GJ* (94.54 ± 144.89), *Aus* (43.58 ± 53.21), *Adm* (29.08 ± 26.97) and *Bas* (21.54 ± 20.42). Notably, the number of major gcHaps accounted for 1.34% of all detected 10,129 gcHaps, 91.35% were unique (present in one population but absent in others), and these unique gcHaps represented a considerable portion of each population (88.97% for *XI* and over 50.00% for the others) although they were infrequent (Figure 4; Appendix A).

Here, the *E_H_* was used to evaluate the degree of the gene’s genetic diversity among individuals and the average Nei’s genetic identity, *I_Nei_*, was used to assess the extent of the gene’s contribution to the population’s differentiation. *Adm* population shared the highest mean *E_H_* (0.48± 0.25), next to *Bas* (*E_H_* = 0.37 ± 0.30), *Aus* (*E_H_* = 0.34 ± 0.28), *XI* (*E_H_* = 0.29 ± 0.25) and *GJ* (*E_H_* = 0.22 ± 0.24). *I_Nei_*, was highest between *GJ* and *Bas* (0.85), small between *GJ* and *Aus*, *GJ* and *XI*, *Bas* and *XI*, *Bas* and *Aus*, and relatively intermediate between other populations (Figure 4; Appendix A).

### 2.6. Analysis of Cis-Regulatory Elements (CREs) in the Promoters of OsAGCs

Cis-Regulatory Elements (CREs) in promoter regulate gene expression by mediating transcriptional processes, CREs were searched in the 2000 bp sequences upstream of the translation start sites of *OsAGC* genes, and the results were shown in Figure 5. The identified CREs were classified into four groups, the first group is abiotic-stress-related elements, including anoxic specific inducibility responsive elements (ARE, GC-motif), drought-responsive elements (MBS), low-temperature responsive elements (LTR), wound responsive element (WUN-motif), defense and stress-responsive element (TC-rich repeats) and abundant light-responsive elements (ACE, Sp1, AT-box, etc.). The second group is hormone-response-related elements, consisting of abscisic acid-induced element (ABRE), jasmonic acid-induced element (CGTCA-motif), salicylic acid-induced element (TCA-element), auxin-induced element (AuxRR-core) and gibberellin induced element (GARE-motif). The third group is development-related elements, containing meristematic tissue expression responsive element (CAT-box), endosperm expression responsive element (GCN4-motif) and differentiation of the palisade mesophyll cells responsive element (HD-Zip 1). The remaining circadian regulatory element (circadian) and zein metabolism regulatory element (O_2_-site) were classified as other groups.

*OsAGC* genes possessed a large number of light-responsive elements, jasmonic acid responsive elements, abscisic acid responsive elements and anoxic specific inducibility responsive elements, among these genes *OsAGC8*, *OsAGC13* and *OsAGC18* shared more light-responsive elements, *OsAGC2* and *OsAGC10* held more jasmonic acid-responsive elements, *OsAGC13*, *OsAGC13* and *OsAGC25* owned more abscisic acid-responsive elements, and *OsAGC15*, *OsAGC16* and *OsAGC26* carried more anoxic specific inducibility responsive elements. Moreover, all members of the *OsAGC* gene family were found to contain light-responsive and anoxic-specific inducibility-responsive elements. In addition, meristem expression responsive elements were identified in more than half of the members. These observations suggested that the *OsAGC* genes may be regulated by various stress responses and developmental processes.

### 2.7. Tissue-Specific Expression Analysis of OsAGC Genes

The occurrence of duplication events caused diversity of gene expression and some specific evolutionary patterns of these genes that meet the needs of plant growth and development, and some of the duplicated genes identified in this study exhibited this variation (Figure 6; Appendix A). The duplicated gene pairs *OsAGC4/OsAGC10* from subfamily I and *OsAGC2/OsAGC12* from subfamily VI were under-expressed in all tissues and they experienced purifying selection, revealing that some genes in these subfamilies may exhibit functional redundancy. The duplicated gene *OsAGC17* was expressed at a higher level in all tissues compared to *OsAGC15*, *OsAGC21*’s expression differed from *OsAGC23* in leaf sheath and pistil, *OsAGC5*’s expression differed from *OsAGC11* in leaf blade and endosperm, the distinctions of expression patterns indicated that these genes may have experienced functional differentiation during evolution. In addition, *OsAGC3*, *OsAGC8*, *OsAGC11* and *OsAGC14* showed high expression levels in all tissues, implying that they are essential for the growth and development of multiple organs. Notably, all members in subfamily V, namely *OsAGC9*, *OsAGC20*, and *OsAGC22*, were highly expressed in leaf and leaf sheath, suggesting that genes in this subfamily may perform the important function in leaf development. In conclusion, the specific expression patterns exhibited by *OsAGC* genes in different tissues demonstrated that gene function may have evolved in response to various environmental processes.

### 2.8. OsAGCs That May Function in Rice Photosynthesis

Public data from RNA-seq indicated that all members of subfamily V were highly expressed at the leaf level; this finding encouraged us to explore the functions of these members in leaf development, and expression patterns of three *OsAGC* genes were analyzed at 47, 50, 53 and 56 days after sowing using qRT-PCR technology (Figure 7). The light-induced genes *OsRL3*, *LC7*, and *DGP1* have been demonstrated to be responsive to chlorophyll expression and photosynthesis in rice, so these three genes were used as reference genes to compare their expression patterns with those of *OsAGCs* [37,38,39]. All three genes were expressed at the measured stages, and the expression patterns of *OsAGC9*, *OsAGC20*, and *OsAGC22* were most similar to those of *OsRL3*, with similar expression levels and expression change trends at all stages. The expression patterns of *OsAGC9*, *OsAGC20*, and *OsAGC22* were also similar to those of *LC7*, their expression peaks and lows occurred at the same stage, and their expressions changed simultaneously, except that the degree of expression change of the three genes was higher than that of *LC7* at 56 DAS. *OsAGC3* was located at the core of the protein–protein interaction network (Appendix A), showing high expression in all tissues, possibly involved in complex physiological processes. The expression of *OsAGC3* in leaves was examined and the expression pattern of this gene was found to be close to that of *DGP1*, with the identical stages of expression changes, although the expression peak of the former appeared at 50 DAS and that of the latter at 56 DAS. In summarization, the above findings indicated that *OsAGC3*, *OsAGC9*, *OsAGC20* and *OsAGC22* may participate in photosynthesis at the leaf level.

## 3. Discussion

Rice has been widely studied as a model plant, the genome annotation information became more detailed with the in-depth sequencing of the genome, which made it possible to identify vital genes of growth and development [40]. In recent years, some gene families, such as *GH3*, *CDCP*, and *SPL* were identified in rice [41,42,43]. The AGC protein kinase family is a subfamily of the protein kinase superfamily that is widely engaged in plant growth regulation and development [4,5]. The first identification of the *AGC* gene family was performed in *A. thaliana*, and numerous functional studies were carried out, confirming that these AGC protein kinases were involved in lipid signaling pathways, auxin regulation, cell proliferation, etc. [6]. There were few functional studies of *AGC* genes in *S. lycopersicum*, *T. aestivuml*, *M. truncatula* and *P. yezoensis*, which were concerned with apoptosis, defense response and cell division, although *AGC* genes in these plants have not been systematically identified [24,25,26,27,28]. To date, the distributions of the *AGC* genes in rice are unknown, and only few genes’ functions were explored, they participated in auxin signal transduction and organ development [31,32,33,34,35,36]. The present study systematically identified the *AGC* genes in rice, and analysis of characterization, sequence structure, motif, collinearity, cis-acting element, selection pressure and haplotype were performed. Moreover, the expression profiles of *AGC* genes were analyzed by combining RNA-seq and qRT-PCR to support a further comprehensive understanding of their function.

Comparative genomic analysis among closely related species can greatly improve our comprehension of gene evolution. This research focused on AA genome *Oryza* species, 26 AGC members were identified in *O. sativa*, *japonica*, 27 members in *O. sativa*, *indica*, 26 members in *O. nivara*, 27 members in *O. refipugon*, 25 members in *O. glaberrima*, 26 members in *O. meridionalis*, 26 members in *O. barthii*, 28 members in *O. glumaepatula*, 29 members in *O. longistaminata* and apparently a comparable number of members were identified in these nine species. The AGC members in the AA genome *Oryza* genus were classified into six subfamilies with reference to the classification of *A. thaliana*, and the number of subfamily members was found to be similar in the nine varieties, indicating that the *AGC* gene family may have undergone the same evolutionary pathway in the AA genome *Oryza* genus. It is commonly known that transposition, segmental duplication, WGD and tandem duplication play vital roles in biological evolution [44]. In the present study, all gene pairs experiencing duplication events were generated by segmental or WGD duplication, implying that segmental or WGD duplication may perform an important role in the expansion of the *AGC* gene family in the AA genome *Oryza* genus. Interestingly, one gene in *O. sativa*, *japonica* (*OsAGC17*), four genes in *O. sativa*, *indica* (*BGIOSGA030968*, *BGIOSGA034806*, *BGIOSGA036995*, *BGIOSGA015138*), four genes in *O. nivara* (*ONIVA09G14540*, *ONIVA04G09670*, *ONIVA01G18870*, *ONIVA12G02630*), four genes in *O. rufipogon* (*ORUFI12G03440*, *ORUFI11G03140*, *ORUFI04G12910*, *ORUFI09G15010*), four genes in *O. glaberrima* (*ORGLA11G0030200*, *ORGLA12G0028500*, *ORGLA02G0263300*, *ORGLA04G0087400*), three genes in *O. barthii* (*OBART04G11720*, *OBART12G03020*, *OBART11G03270*), three genes in *O. glumaepatula* (*OGLUM11G02870*, *OGLUM09G14610*, *OGLUM04G20570*) existed in multiple collinear gene pairs suggested that these genes may be irreplaceable in the expansion of *AGC* gene family. The Ka/Ks ratio can be applied to estimate the historical selection of coding sequences, Ka/Ks analysis in this study showed that all duplicated *AGC* gene pairs in the AA genome *Oryza* genus experienced disadvantageous selection, demonstrating that elimination of unfavorable mutations may facilitate rice to adapt to complicated environments.

Photosynthesis is the physiological basis for the high production of plants, and leaves are the key organ of photosynthesis, and chloroplasts in leaves are the location of photosynthesis [45]. *AtPHOT1* and *AtPHOT2* in the AGC protein kinase subfamily V act as blue light photoreceptors in the signal-transduction pathway for photo-induced movements, promoting stomatal opening and chloroplast accumulation [22,23]. Moreover, these two genes were found to have high expression in leaves in the Expression Atlas database [46], and the rice genes *OsAGC9*, *OsAGC20*, *OsAGC22* in this subfamily also showed high expression in leaves in the RiceXPro database [47], indicated that the genes of subfamily V in rice and *Arabidopsis* may have the similar functions and are essential for leaf development. Genes with close expression patterns could perform the same functions. The expression patterns of genes that were highly expressed in leaves were compared with those of validated photosynthesis-related genes (*OsRL3*, *LC7*, *DGP1*) in rice by qRT-PCR, and found that the expression patterns of *OsAGC9*, *OsAGC20*, *OsAGC22* were similar to those of *OsRL3* and *LC7*. *OsAGC3* located at the core of the protein-protein interaction network shared a close expression pattern with *DGP1*, suggesting that *OsAGC3*, *OsAGC9*, *OsAGC20*, *OsAGC22* may be responsible for photosynthesis at the leaf level. In addition, protein-protein interaction prediction revealed no reciprocal relationship between these four genes (Appendix A), implying that they may execute function through their respective pathways.

## 4. Materials and Methods

### 4.1. Identification of AGC Genes from Rice

Twenty AGC protein sequences of *Arabidopsis* in the UniProtKB/Swiss-Prot (SwissProt) database (https://www.uniprot.org, accessed on 1 June 2022) were used to construct the Hidden Markov model (HMM) file of the AGC domain and searched in the genome of *O. sativa*, *japonica*, *O. sativa*, *indica*, *O. nivara*, *O. rufipogon*, *O. glaberrima*, *O. meridionalis*, *O. barthii*, *O. glumaepatula* and *O. longistaminata* from Ensembl Plants database (http://plants.ensembl.org, accessed on 1 June 2022). Meanwhile, twenty AGC protein sequences were used as queries to perform protein blast (blastp) search in the genomes of nine *Oryza* species. All the identified sequences were tested for the existence of AGC domain in the HMMER tool (https://www.ebi.ac.uk/Tools/hmmer/search/hmmscan, accessed on 1 June 2022) [48]. All accession names of *AGC* genes were shown in Appendix A.

### 4.2. Characteristics, Gene Structure, Motif and Phylogenetic Relationship Analysis of AGC Gene Family

The protein parameter calc tool in TBtools was used to predict the number of amino acids, molecular weight (MW) and isoelectric point (pI) of proteins, and the simple MEME wrapper tool was employed to identify motifs, and gene structure was visualized with this software [49]. BUSCA (http://busca.biocomp.unibo.it, accessed on 1 June 2022) was utilized to analyze the subcellular localization of AGC proteins [50]. MEGA X was used to perform multiple sequence alignments and construct phylogenetic trees [51]. Sequences were aligned according to the method ClustalW (default parameters) and evolutionary trees were constructed based on the maximum likelihood method (default parameters). The results were visualized using TBtools and iTOL (https://itol.embl.de, accessed on 1 June 2022) [52].

### 4.3. Collinearity Analysis of AGC Genes

The collinearity analysis was performed using the MCScanX toolkit with default parameters [53] to explore the duplication events of *AGCs* in AA genome rice species, and the distribution of collinear pairs was presented by TBtools. The simple Ka/Ks calculator tool of TBtools was applied to calculate Ka, Ks and Ka/Ks ratios of duplicated genes. Divergence time (T) is calculated by the equation T = Ks/(2 × 9.1 × 10^−9^) × 10^−6^ million years ago (Mya) [54].

### 4.4. gcHap Analysis of the AGC Genes in 3KRG

The gcHap data of *AGCs* were obtained from the RFGB database (https://www.rmbreeding.cn, accessed on 1 June 2022) [55], and analyzed using the Personalbio Cloud online platform (https://cloud.metware.cn, accessed on 1 June 2022).

### 4.5. Identification of Cis-Regulatory Elements (CREs)

Retrieve the 2 kb upstream promoter sequences of the transcription start sites of the *AGC* genes of *O. sativa*, *japonica* from the Ensembl Plants database (http://plants.ensembl.org, accessed on 1 June 2022), and searched cis-acting elements of these genes using the PlantCARE online tool (http://bioinformatics.psb.ugent.be/webtools/plantcare/html/, accessed on 1 June 2022) [56]. The information on CREs was shown in Appendix A.

### 4.6. Expression and Interaction Analysis of OsAGC Gene Family

Data on the expression profile of the *AGC* genes in various tissues at different developmental stages were retrieved from the RiceXPro database (https://ricexpro.dna.affrc.go.jp, accessed on 1 June 2022) [47] and analyzed using the Metware Cloud online platform (https://cloud.metware.co.uk, accessed on 1 June 2022). Leaves were taken for qRT-PCR on 47, 50, 53 and 56 days after sowing, and all reagents used were from Vazyme, China. Total RNA was isolated using Trizol reagent, cDNA preparation using HiScript III RT SuperMix, and quantitative analysis was performed using ChamQ SYBR qPCR Master Mix. The relative expression levels of *OsAGCs* were determined according to the 2^−∆∆CT^ method, with *OsActin* as an internal reference. The primers’ information was shown in Appendix A.

The STRING database (https://cn.string-db.org, accessed on 1 June 2022) was used to predict the interaction relationship between OsAGC members [57].

## 5. Conclusions

This study filled the gap of information on the AGC protein kinase family in *Oryza sativa* ssp. *japonica* and its closely related species by performing the analysis of gene structure, conserved motifs, phylogeny, collinearity, Ka/Ks and haplotype. We found that the gene family is highly conserved in rice and the expansion was associated with the occurrence of segmental or WGD duplication and purifying selection. Abundant light-responsive elements were detected and some *AGC* genes that possessed similar expression patterns to those of photosynthesis-related genes were found, indicating that the family may function in photosynthesis. Overall, this work broadened the knowledge of the *AGC* gene family and provided useful information for further elaboration of *AGC* genes’ regulatory mechanism in rice.

## Figures and Tables

**Figure 1 ijms-23-12557-f001:**
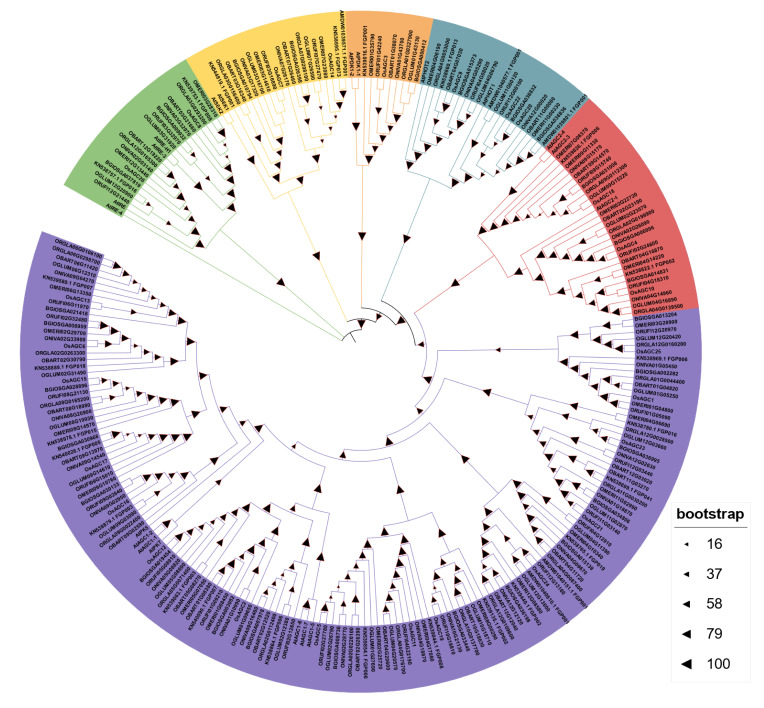
Phylogenetic relationship of AGC proteins identified in *Oryza* species and *Arabidopsis*.

**Figure 2 ijms-23-12557-f002:**
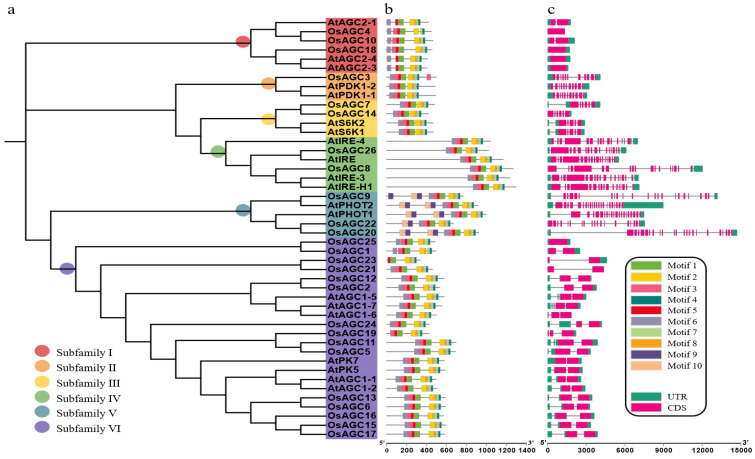
Phylogenetic relationship, gene structure and conserved motifs of *AGC* family members. (**a**) Phylogenetic relationship of AGCs in rice and *Arabidopsis*. (**b**) Structure of exons and introns of *AGCs*, exons and introns were represented using rectangles and bars, respectively. (**c**) Distributions of conserved motifs.

**Figure 3 ijms-23-12557-f003:**
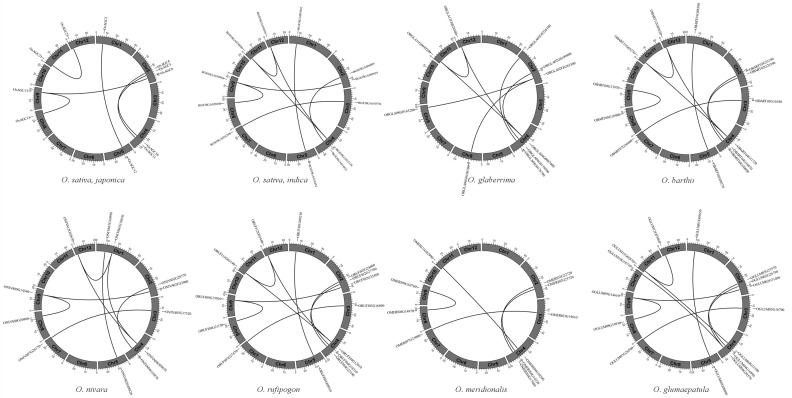
Collinearity analysis of *AGCs* in *Oryza* species.

**Figure 4 ijms-23-12557-f004:**
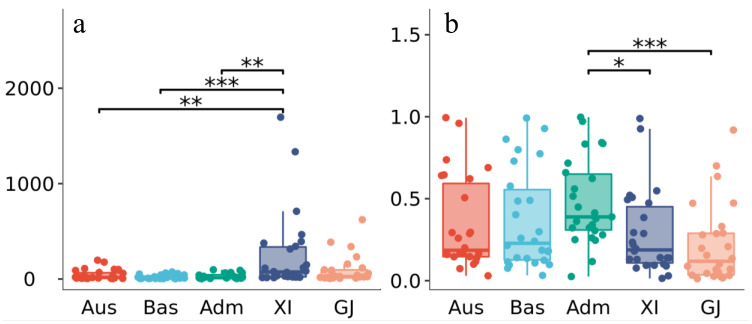
Haplotype analysis of the *AGC* gene family in 3010 rice genomes (3KRG). (**a**) Distribution of gene–CDS–haplotype number (gcHapN) in different populations. (**b**) Distribution of equitability values (*E_H_*) in different populations. *Bas*, *Basmati*; *Adm*, *Admixtures*; *XI*, *Xian*/*Indica*; *GJ*, *Geng*/*Japonica*; Different colors represent different populations; * Difference is significant at *p* < 0.05 level among different populations; ** Difference is significant at *p* < 0.01 level among different populations; *** Difference is significant at *p* < 0.001 level among different populations.

**Figure 5 ijms-23-12557-f005:**
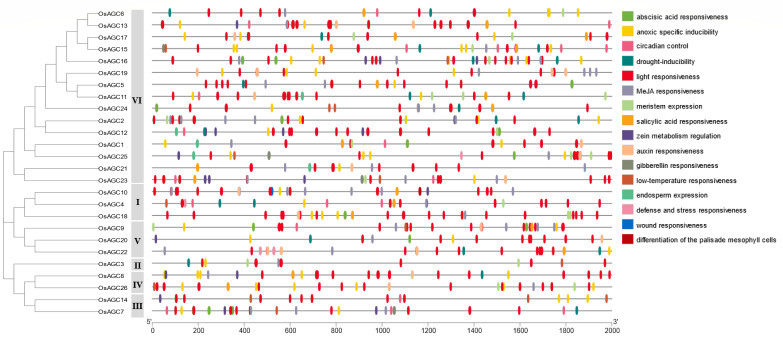
Predicted cis-elements of the *OsAGC* gene family.

**Figure 6 ijms-23-12557-f006:**
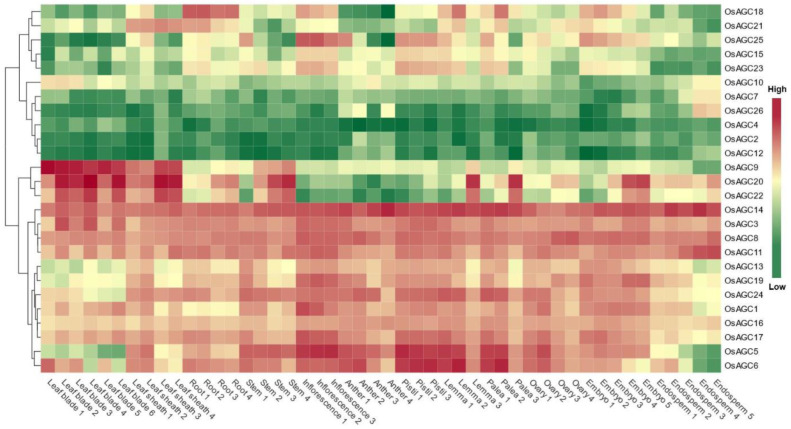
Expression patterns analysis of *OsAGCs* in *O. sativa*, *japonica*.

**Figure 7 ijms-23-12557-f007:**
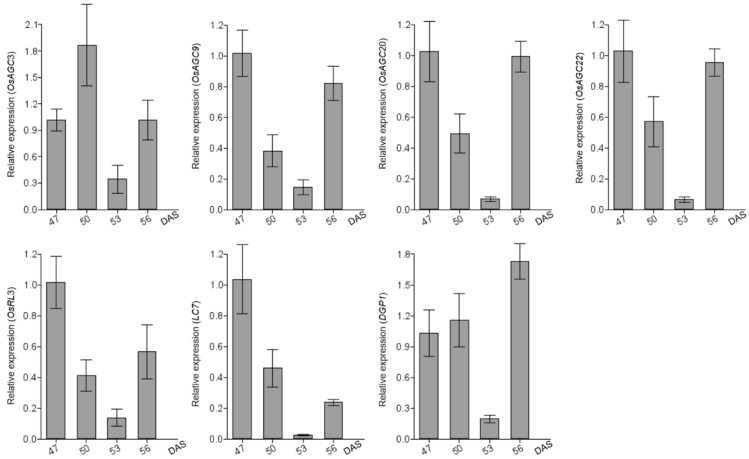
The expression profiles of four *OsAGCs* and three photosynthesis-related genes were evaluated by qRT–PCR in leaf-level at 47, 50, 53, and 56 days after sowing (DAS).

**Table 1 ijms-23-12557-t001:** Basic information of *OsAGC* family genes and their proteins in *O. sativa*, *japonica*.

Gene	Locus ID	Chr	Size (aa)	MW(Da)	pI	Subcellular Location
*OsAGC1*	Os01g0174700	1	493	54,366	8.86	Nucleus
*OsAGC2*	Os01g0233800	1	532	57,906	9.14	Nucleus
*OsAGC3*	Os01g0872800	1	498	55,384	6.54	Nucleus
*OsAGC4*	Os02g0603000	2	447	47,166	8.37	Nucleus
*OsAGC5*	Os02g0654300	2	690	74,999	6.15	endomembrane
*OsAGC6*	Os02g0725000	2	588	65,088	8.06	Nucleus
*OsAGC7*	Os03g0334000	3	480	53,316	6.29	Nucleus
*OsAGC8*	Os03g0711800	3	1267	138,833	5.94	endomembrane
*OsAGC9*	Os04g0304200	4	771	86,407	8.46	Nucleus
*OsAGC10*	Os04g0488700	4	462	49,336	5.65	Nucleus
*OsAGC11*	Os04g0546300	4	695	75,638	6.38	Nucleus
*OsAGC12*	Os05g0237400	5	574	61,434	7.00	Nucleus
*OsAGC13*	Os06g0291600	6	589	64,808	8.38	chloroplast
*OsAGC14*	Os07g0680900	7	419	46,867	7.87	Nucleus
*OsAGC15*	Os08g0491200	8	594	65,034	8.74	plasma membrane
*OsAGC16*	Os09g0258500	9	567	61,832	8.21	Nucleus
*OsAGC17*	Os09g0478500	9	583	63,813	7.18	Nucleus
*OsAGC18*	Os09g0486700	9	455	49,179	6.34	plasma membrane
*OsAGC19*	Os10g0562500	10	426	46,678	9.21	Nucleus
*OsAGC20*	Os11g0102200	11	921	103,437	8.35	Nucleus
*OsAGC21*	Os11g0150700	11	458	49,130	6.43	Nucleus
*OsAGC22*	Os12g0101800	12	668	76,000	8.31	chloroplast
*OsAGC23*	Os12g0149700	12	338	36,565	6.28	endomembrane
*OsAGC24*	Os12g0480200	12	430	48,355	6.39	Nucleus
*OsAGC25*	Os12g0614600	12	484	51,769	9.50	chloroplast outer membrane
*OsAGC26*	Os12g0621500	12	1021	113,490	5.32	Nucleus

Chr, chromosome; MW, molecular weight; pI, isoelectric point.

**Table 2 ijms-23-12557-t002:** The Ka/Ks values in duplicated gene pairs in *Oryza* species.

*Oryza* Species	Gene Pairs	Ka	Ks	Ka/Ks	Date (Mya)	Type of Selection	Type of Duplication
*O. sativa*, *japonica*	*OsAGC2/OsAGC12*	0.23	0.66	0.35	36.26	Purifying	Segmental/WGD
	*OsAGC4/OsAGC10*	0.21	0.46	0.45	25.44	Purifying	Segmental/WGD
	*OsAGC5/OsAGC11*	0.12	1.08	0.11	59.56	Purifying	Segmental/WGD
	*OsAGC15/OsAGC17*	0.10	0.96	0.10	52.91	Purifying	Segmental/WGD
	*OsAGC21/OsAGC23*	0.05	0.17	0.27	9.34	Purifying	Segmental/WGD
*O. sativa*, *indica*	BGIOSGA003082/BGIOSGA019454	0.23	0.65	0.34	35.93	Purifying	Segmental/WGD
	BGIOSGA034806/BGIOSGA036995	0.08	0.20	0.40	10.93	Purifying	Segmental/WGD
	BGIOSGA006096/BGIOSGA014831	0.21	0.46	0.45	25.44	Purifying	Segmental/WGD
	BGIOSGA010794/BGIOSGA026398	0.14	1.13	0.12	62.25	Purifying	Segmental/WGD
	BGIOSGA028896/BGIOSGA030968	0.09	0.97	0.09	53.19	Purifying	Segmental/WGD
*O. nivara*	ONIVA01G18870/ONIVA12G02630	0.05	0.22	0.25	12.25	Purifying	Segmental/WGD
	ONIVA01G10990/ONIVA05G08820	0.22	0.64	0.34	35.39	Purifying	Segmental/WGD
	ONIVA02G28770/ONIVA04G18870	0.12	1.08	0.11	59.34	Purifying	Segmental/WGD
	ONIVA03G17320/ONIVA07G26170	0.11	1.06	0.10	58.30	Purifying	Segmental/WGD
	ONIVA08G20900/ONIVA09G14540	0.09	0.97	0.10	53.08	Purifying	Segmental/WGD
*O. rufipogon*	ORUFI01G09210/ORUFI05G08910	0.23	0.63	0.37	34.56	Purifying	Segmental/WGD
	ORUFI11G03140/ORUFI12G03440	0.05	0.24	0.22	13.35	Purifying	Segmental/WGD
	ORUFI02G24600/ORUFI04G18310	0.21	0.46	0.44	25.44	Purifying	Segmental/WGD
	ORUFI02G27780/ORUFI04G22190	0.12	1.08	0.11	59.45	Purifying	Segmental/WGD
	ORUFI03G16890/ORUFI07G27470	0.11	1.04	0.10	57.20	Purifying	Segmental/WGD
	ORUFI08G21130/ORUFI09G15010	0.10	0.97	0.10	53.24	Purifying	Segmental/WGD
*O. glaberrima*	ORGLA11G0030200/ORGLA12G0028500	0.06	0.23	0.28	12.63	Purifying	Segmental/WGD
	ORGLA02G0226100/ORGLA04G0176700	0.12	1.08	0.11	59.09	Purifying	Segmental/WGD
	ORGLA02G0199900/ORGLA04G0139500	0.20	0.47	0.43	25.72	Purifying	Segmental/WGD
	ORGLA02G0263300/ORGLA06G0106100	0.13	1.34	0.10	73.37	Purifying	Segmental/WGD
	ORGLA02G0263300/ORGLA08G0165200	0.18	2.50	0.07	137.23	Purifying	Segmental/WGD
*O. meridionalis*	OMERI02G25720/OMERI04G17080	0.12	1.07	0.12	58.74	Purifying	Segmental/WGD
	OMERI02G22720/OMERI04G14220	0.22	0.48	0.46	26.52	Purifying	Segmental/WGD
	OMERI03G14610/OMERI07G23080	0.11	1.01	0.11	55.25	Purifying	Segmental/WGD
	OMERI08G14970/OMERI09G10760	0.092	0.97	0.09	53.40	Purifying	Segmental/WGD
*O. barthii*	OBART01G08380/OBART05G08270	0.30	0.72	0.41	39.31	Purifying	Segmental/WGD
	OBART11G03270/OBART12G03020	0.09	0.23	0.38	12.69	Purifying	Segmental/WGD
	OBART02G23190/OBART04G16870	0.35	0.51	0.70	27.86	Purifying	Segmental/WGD
	OBART02G26390/OBART04G20600	0.12	1.08	0.11	59.09	Purifying	Segmental/WGD
	OBART03G16340/OBART07G26460	0.11	1.06	0.10	58.29	Purifying	Segmental/WGD
	OBART08G18980/OBART09G13970	0.09	0.96	0.10	53.02	Purifying	Segmental/WGD
	OBART08G19720/OBART09G14570	0.20	0.47	0.44	25.11	Purifying	Segmental/WGD
*O. glumaepatula*	OGLUM01G09650/OGLUM05G08690	0.22	0.62	0.35	34.34	Purifying	Segmental/WGD
	OGLUM11G02870/OGLUM12G03660	0.06	0.22	0.26	12.23	Purifying	Segmental/WGD
	OGLUM12G20420/OGLUM03G27960	0.20	0.54	0.38	29.76	Purifying	Segmental/WGD
	OGLUM02G23570/OGLUM04G16890	0.20	0.46	0.44	25.18	Purifying	Segmental/WGD
	OGLUM02G26790/OGLUM04G20570	0.12	1.08	0.11	59.49	Purifying	Segmental/WGD
	OGLUM03G16700/OGLUM07G26500	0.11	1.07	0.10	59.05	Purifying	Segmental/WGD
	OGLUM08G19930/OGLUM09G14610	0.10	0.96	0.10	52.78	Purifying	Segmental/WGD
	OGLUM08G20880/OGLUM09G15220	0.21	0.42	0.50	23.32	Purifying	Segmental/WGD

Ka, the ratio of non-synonymous substitution; Ks, the ratio of synonymous substitution; Mya, million years ago.

## Data Availability

Data is contained within the article or Appendix A.

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
