# Peer review of "Genome-Wide Identification of the AGC Protein Kinase Gene Family Related to Photosynthesis in Rice (Oryza sativa)"

_ijms, 2022, doi:10.3390/ijms232012557_

Round 1
Reviewer 1 Report
The manuscript entitled “Genome-Wide Identification of the AGC Protein Kinase Gene 2 Family Related to Photosynthesis in Rice” is focused on the role of ACG gene family in rice. The manuscript falls within the scope and contains useful information for the readers of IJMS. The idea of this manuscript is interesting, most notably because the authors have used different lines of rice, including wild rice. The analysis is good, and scientific contents are also good. However, the manuscript was not proof read before submission and it contains few syntax and grammar errors. The following are the specific comments and suggestions:
· Authors should proofread whole manuscript carefully to remove type/linguistic errors. There are several mistakes in Abstract, such as L9 in-volving, L16 in-dicated, L18 re-vealed , L20 photosynthe-sis-related .
· Moreover, all scientific names should be italic such as Oryza sativa, indica, japonica etc. Please change in whole manuscript
· L9-10 Please re-phrase
· Gene names format should be same in whole manuscript, better to use as italic
· Table 1: it should be size, not siza. sometime nucleus first character mentioned as upper case letter while sometime as small. Please be consistent
· L120 please change it as "and O. longistaminata, respectively.
· L120-125, Need to re=phrase
· Fig. 2: Figures should be self-explanatory. Different colors indicate what? plz add legends.
· Please use full name for any abbreviation at first instance, such as L135, WGD
Reviewer 2 Report
The authors performed systematic study on ACG gene family in rice. I have some queries as listed bellow.
Please add the scientific name in the title.
Line 8: Please don’t start with small letter. Add “The” before starting the sentence.
Line 11-15: All the scientific names should be in italic.
Line 19-20; Line 51: The name of genes should be in italic. Please check through out manuscript.
Line 36-38; 52: All the scientific names should be in italic. Please check through out manuscript.
Section 2.2: Please add about intron-exons in this section. This is missing.
Please add the bootsharp in Fig-2.
Please delete the additional line highlighted In the Table-2.
2.3 will come first as 2.1 followed by other sub heads. Please arrange accordingly.
The authors performed RT PCR analysis with 7 genes. How they selected these 7 genes for expression analysis; should be added in M7M section.
In reference, please correct the numbering.
Round 2
Reviewer 2 Report
Thanks to the authors for incorporation of comments for improvement of the manuscript. But still some minor corrections are there, please try to incorporate.
1. Fig-1: Arabidopsis will be in italic.
2. Table-1: In subcellular localization: Capitalize each starting letter as “Nucleus”.
3. Line 335: The name of genes will be in italic.
4. Line 407: subsp. Is ok.
5. In reference: Please check that all the scientific names will be wrote in italics. I see the errors at several places in this section.
6. Line 452: The journal name will be in italic. Please go through the reference section and correct it.
Author Response
Dear Reviewer 2:
Thanks for your comments concerning our manuscript, these comments are all valuable and very helpful for improving our manuscript. We have made correction according to these comments, and we hope the revised manuscript would meet with approval. The revisions are marked up using the “Track Changes” function in the paper. The point-to-point responds to your comments are as following:
Point 1: Fig-1: Arabidopsis will be in italic.
Response 1: Thanks for your comment, we have revised the format of word ‘Arabidopsis’ in whole manuscript.
Point 2: Table-1: In subcellular localization: Capitalize each starting letter as “Nucleus”.
Response 2: Thanks for your comment, we have added.
Point 3: Line 335: The name of genes will be in italic.
Response 3: Thanks for your comment, we have revised.
Point 4: Line 407: subsp. Is ok.
Response 4: Thanks for your comment, we used ssp in whole manuscript.
Point 5: In reference: Please check that all the scientific names will be wrote in italics. I see the errors at several places in this section.
Response 5: Thanks for your comment, we have revised the formats of all the scientific names in this section.
Point 6: Line 452: The journal name will be in italic. Please go through the reference section and correct it.
Response 6: Thanks for your comment, we have corrected the formats of journal names in this section.